# Insight Discovery of the Roman Amphitheater of Durres: Reconstruction of the Acoustic Features to Its Original Shape

Antonella Bevilacqua [1,*] , Silvana Sukaj [2] , Gino Iannace [3] and Amelia Trematerra [3]

1 Department of Industrial Engineering, University of Parma, 43124 Parma, Italy
2 Department of Engineering and Architecture, European University of Tirana (UET), 1000 Tirana, Albania; silvana.sukaj@uet.edu.al
3 Department of Architecture and Industrial Design, University of Campania "Luigi Vanvitelli", 81031 Aversa, Italy; gino.iannace@unicampania.it (G.I.); amelia.trematerra@unicampania.it (A.T.)
* Correspondence: antonella.bevilacqua@unipr.it

**Abstract:** The discovery of Roman amphitheaters continues to excite the minds of archaeologists. Within the framework of various excavation campaigns, the architectural requalification of the amphitheater of Durres has been investigated, but no acoustic analyses have yet been carried out. This paper deals with the acoustic reconstruction of the Roman amphitheater of Durres in its original form. A campaign of acoustic measurements was carried out in accordance with ISO 3382 in order to understand the existing conditions, which are very detrimental to any type of live performance. After an accurate analysis of the geometric composition of the building, acoustic simulations were performed to determine the original acoustic response of the building. A comparison of the measured and simulated results, with and without an audience, was made in terms of the main acoustic parameters, while the acoustic map showing the spatial distribution of speech clarity at 1 kHz was added as an alternative way of representing the data. The outcomes show that the results related to the original shape are closer to the optimal values than the existing conditions.

**Keywords:** Roman amphitheaters; open-air amphitheaters; acoustic simulations; acoustic measurements; cultural heritage





## 1. Introduction

Roman amphitheaters were considered the largest buildings in the Roman world. The Greek name *amphi*, added to the noun *teatrum*, does not refer to a double theater but, more precisely, to a place where it was possible to attend a live event standing around the stage, regardless of which side.

This paper deals with the acoustics of the amphitheater of Dyrrachium (Roman name for Durres), as virtually reconstructed. The work was based on previous historical research on Roman amphitheaters. For this particular case, a three-dimensional geometric reconstruction of the amphitheater was performed starting from the existing conditions of the site, which only some parts of the *cavea* and arena remain. Acoustic measurements were carried out across the archaeological site to gather the current acoustic characteristics. From the measured values, a virtual model was reconstructed in order to be analyzed with software suitable for architectural acoustics. The acoustic simulations carried out with Ramsete software in two specific scenarios of the original shape: with and without audience. The outcomes of this study would be useful for virtual reality or cinematographic business when the environments mirror ancient periods.

The amphitheater was used for two types of spectacles: *munera*, which consisted of combats among gladiators, and *venationes* in which gladiators competed against wild beasts. The first structures in stone were realized in Campania, specifically in Pompeii, Capua, Cuma and Pozzuoli; from there, this construction typology was replicated in all cities of the Roman Empire. The realization of an amphitheater in a city had a political function,

since governors encouraged citizens to participate in the life of the empire by attending the spectacles to increase the worship of the emperor [1].

The first permanent amphitheater was built in Pompeii in 70 BC. The Campanian amphitheater in Capua, the place where the gladiator Spartacus led his revolt against Rome, was used as a model for the construction of the Colosseum in Rome, which is the most famous amphitheater in the world. In Italy there is also the amphitheater of Verona actively used for live performance, like opera and other entertainment activities and results the best preserved.

The amphitheaters were built with stone blocks and terracotta bricks, the *cavea* with its steps rested on a series of multiple arches to accommodate the structure to the uneven natural ground. The arch is a typical Roman construction which is the primitive geometry of complex volumes.

The geometrical description of how the amphitheaters were built was not reported in any manual, except for Book V of *De Architectura*, written by Lucius Vitruvius Pollio [2], in paragraph 6, line 1 of the book, describing the theatrical architecture and the geometrical instructions that should be followed for a proper building. The reasons why plan of amphitheaters is neither circular (given by two mirrored theaters) nor squared are not clarified by archaeologists. It is known that the first combats took place inside the *forum* (with a squared geometry), and it can be assumed that the amphitheaters are the result of a progressive adaptation of existing spaces. Although the reasons may be many, the purpose of their construction was to accommodate the maximum number of spectators and to provide an adequate view of the stage.

Many researchers who have studied the geometry of Roman amphitheaters have debated whether it was an ellipse or an oval [3]. The main difference between the two curves is that the oval is formed by arcs of different radii that cross at certain points, where the arcs have the same tangent, while an ellipse is a closed curve, where the sum of the distances from the two focal points of the major axis is constant. Considering that very simple construction instruments were available in the Roman period (for example, poles and strings to draw onto the soil), it can be deduced that the simplest and quickest geometry to be realized was an oval provided with four centers: two centers on the major axis and the other two on the minor axis [4].

In terms of architectural composition, the amphitheater consists of an elliptical/oval arena, corresponding to the stage floor on which the action took place. The name arena comes from the sand (*arena*) used to cover the wooden planks installed above the machinery of the underground backstage [5]. The other important architectural element is the seating area (*cavea*) where the audience was seated during the performance. The *cavea* was provided with steps, interrupted by annular corridors called *praecintiones*. Access to the *cavea* was through a very efficient system of stairwells and galleries that allowed the audience to enter and leave in a relatively short time. Some amphitheaters were provided with a *velarium*, a system mounted at the top through a cable frame to protect them from overheating, especially during summer seasons [6]. Inside the amphitheaters, through the openings (*vomitoria*), spectators could reach their seat in the *cavea*.

The tradition of spectacles in Roman amphitheaters has become known to the modern age through the narratives of the poets and philosophers of ancient Rome but also through paintings, mosaics and statues found via archaeological excavations in every city.

The growth of Christianity in the 4th century, based on ideologies that rejected the cruel spectacles, such as hunts and gladiatorial combats, led to the abandonment of these buildings. The economic crisis of the Middle Ages took advantage of this situation and favored the reuse of the building materials of theaters and amphitheaters for new projects. Therefore, Roman buildings were demolished or heavily modified by the construction of residences within the ancient structures [5].

Nowadays, Roman amphitheaters are open to visitors for museum purposes, and others, better preserved, are used for live musical performances. The idea by local authorities to put them into operation attracted the attention of researchers, who began to scientifically study the amphitheaters from an acoustic perspective. According to the literature, acoustic

measurements inside the Roman amphitheater of Pompeii show that strong reflections come from the *balteum* of the *cavea* [6], which is the wall separating the steps of the *cavea* from the level of the arena, while other studies focused on the acoustic improvements that can be achieved by adding acoustic panels and/or shells to allow for live musical performances for modern use [7]. Other research studies focused on the acoustic reconstruction of Roman theaters and amphitheaters based on numerical models that reflect the original shapes of these ancient buildings [8].

## 2. Historical Background

The amphitheater of *Dyrrhachium* (the Roman name for Durres) is one of the many surviving Roman structures on the Balkan peninsula and, as far as known, the only one in Albania. It is located in the west of the ancient city of Durres, in a peripheral area of the urban center, near the newer Byzantine walls [9].

According to the archaeological findings, the amphitheater of Durres may have been built between 98 and 117 during the reign of Emperor Trajan. In the same period, a library was also built in the city with the intention of creating an entertainment and cultural center in this part of the Roman city. The first phase of the amphitheater ended in 286, when *Dyrrhachium* was included in a new administrative division by Diocletian [10]. The city was in a strategic position and developed trade with the Adriatic coast and Asia.

A severe earthquake in 346 heavily damaged the amphitheater, which was later worsened by its abandonment in the second half of the 4th century, when the gladiators' exhibitions were banned by Emperor Theodosius for excessive cruelty, which was considered immoral [9]. As such, the architectural elements of the amphitheater (e.g., columns, capitals and fine decorations) were reused for other constructions.

In the 5th and 6th centuries, the Byzantine fortress walls, with a total length of 4400 m and a height of 12 m, were built around the amphitheater, along its outer perimeter. The disuse of the amphitheater continued in the 7th century, when part of the arena and galleries were converted into a Christian cemetery (*necropolis*) [9]. In the following centuries, between the 7th and 10th centuries, residencies and chapels were built, taking advantage of the existing walls. The burial of the amphitheater continued during the Renaissance, when the Turks continued to build residences on the Roman construction [10].

The amphitheater of Durres was discovered only during the second half of the 20th century, more precisely in 1966, when an archaeological excavation campaign was directed by Vangjel Toçi.

Nowadays, the amphitheater is partially visible and settled on the hill, but a large area is buried or destroyed beneath posthumous buildings, as shown in Figure 1.

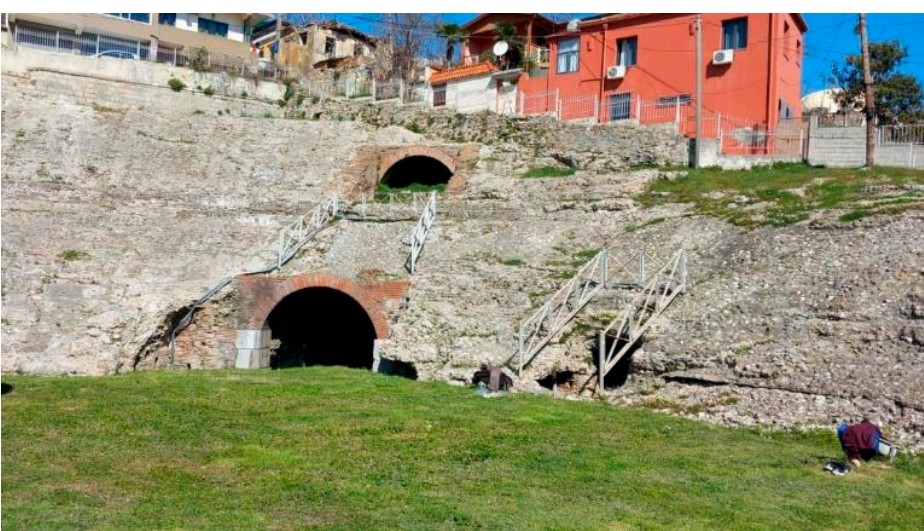

**Figure 1.** Actual view of the Roman amphitheater of Durres.

## 3. Architectural Characteristics of the Existing Conditions

The arena was surrounded by a podium so that the wild animals could not attack the spectators. The *cavea* was divided into two sectors (*maeniana*) by an annular corridor (*praecinctio*) which divided the *ima* from the *summa* cavea [10]. It can be assumed that the amphitheater was provided with a *summa cavea*, but this cannot be confirmed, since no traces have been preserved [11]. The total dimensions of the amphitheater are unclear, since the archaeological excavations have uncovered only part of the arena, while the outer area is still under posthumous constructions or decayed. The intention of digging underground was not executed, since the galleries are obstructed and not safe [11].

Regarding the construction techniques, the *opus coementicium* characterizes the structural walls, sometimes alternating with *opus mixtum* for the brick bands and *opus incertum* for the stone [11]. The limestone of the seats is completely removed, and the structure in *opus coementicium* is visible. In summary, two-thirds of the entire construction has survived to today, as shown in Figure 2.

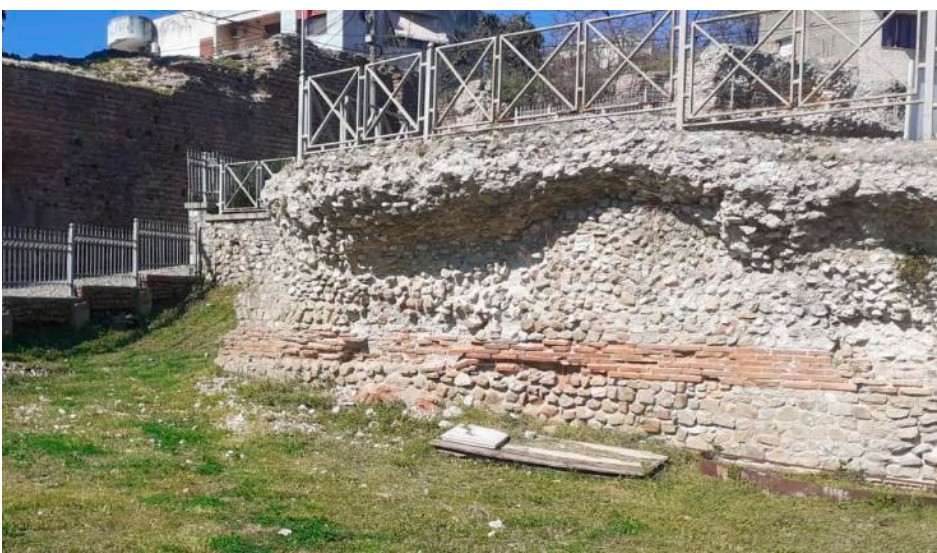

**Figure 2.** View of the wall structure of a gallery belonging to the Roman amphitheater of Durres.

## 4. Attempted Detection of the Roman Geometry

A study of other Roman amphitheaters conducted by the authors led to the identification of a set of parameters that characterize the unique way in which the Romans designed an amphitheater [9]. These parameters are identified as two axes and a curvature that outline the arena, as well as their ratio, the number of arches on the external elevation and the number of wedged sectors that divide the *cavea.*

Based on the visible elements, a polycentric oval is more plausible than an ellipse, since the design of an ellipse in reality is very difficult, although not impossible, which implies a constant distance of the steps with respect to the focal points. In addition, because of a hill on which part of the structure was built, it was easier to trace an oval geometry consisting of a series of circular arcs [12]. In this way, the complexity is increased by the fact that there is only one ellipse with the dimensions of two axes; however, there are an infinite number of oval geometries with axes of the same length but different curvatures. If the theory of the practical realization of an oval with Roman instruments is followed, the complexity can be reduced to a common factor found in many Roman amphitheaters: the geometric construction of the amphitheater is based on a triangle whose vertices correspond to the centers of the radii of the oval [12]. On this basis, many Roman amphitheaters are built on a half equilateral triangle, the "sacred triangle", which forms an oval with four centers.

The usual ratio of the three sides was 3:4:5, upon which Pythagoras's theorem was based [13]. This mathematical ratio determines the harmonic oval that is the geometric baseline of the amphitheater of Durres, as shown in Figure 3.

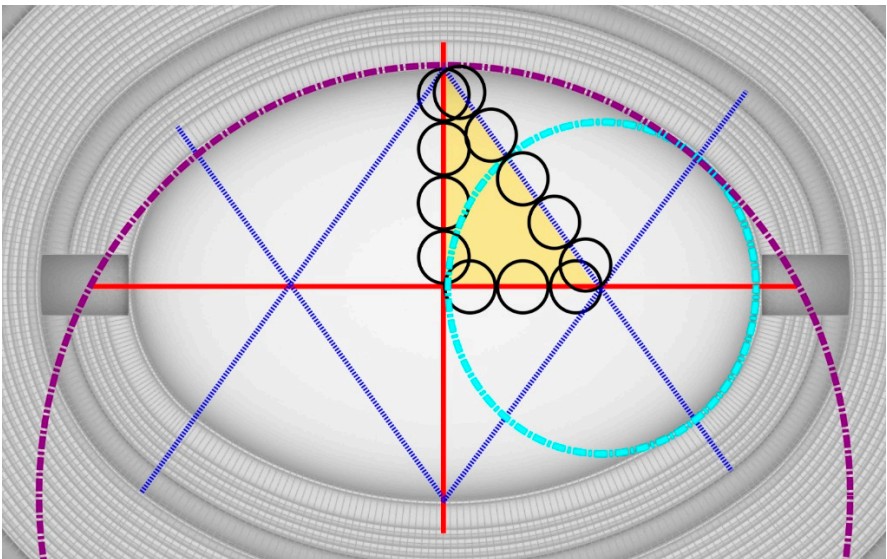

**Figure 3.** Geometry of an oval provided with 4 centers based on a triangle composed using the Pythagorean ratio of 3:4:5.

Regarding the ratio among the axes, the ancient constructions usually used 3/2 or 5/3, although the ratio 3/2 approximates the circumference more than ratio 5/3. In Durres, because of the discrepancy between the geometry of the arena and that of the outer perimeter, the possibility of having an oval with four centers for the tracing of the terrain at the level of the arena and an oval with eight centers for the elevated structures (i.e., ambulatories, steps of the *praecinctio* and external elevation) was studied [13].

Based on this theory, the actual dimensions of the amphitheater of Durres are the following:

- The module is equal to 17 Roman foot (corresponding to 499.5 cm);
- Pure-number dimensional ratio 3:4:5 is equal to 51:68:85 Roman foot;
- Minor axis is equal to 136 Roman foot (corresponding to 8 modules);
- Major axis is equal to 204 Roman foot (corresponding to 12 modules).

### 5. Acoustic Measurements

The acoustic measurements were carried out in the amphitheater of Durres to analyze the acoustic response in the existing conditions. It was not possible to use a loudspeaker due to the absence of the electricity system. Therefore, the following equipment was used:

- Firecrackers;
- Omnidirectional microphone (B&K 4155, 1/2 inch).

The firecrackers have a good signal-to-noise ratio (SNR) [14], although the uncertainties of repeatability and directivity are the main characteristics that can potentially affect the results [15]. Although the excitation signal has a more limited bandwidth spectrum than any other electronic signal (e.g., exponential sine sweep (ESS)), the survey was performed in accordance with the standard requirements outlined by ISO 3382-1 [16]. Measurements were made during the daytime, with an outdoor temperature of 13–15 °C and a wind speed $v < 3.0$ m/s [17].

The location of the sound source was selected to be in the arena, both in the center and off axis, where the firecrackers were ignited at a height of 1.1 m above the ground, and the

microphone was placed at different positions across the *cavea* at a height of 1.3 m, wherever accessibility was allowed. There were five receiving positions, as shown in Figure 4.

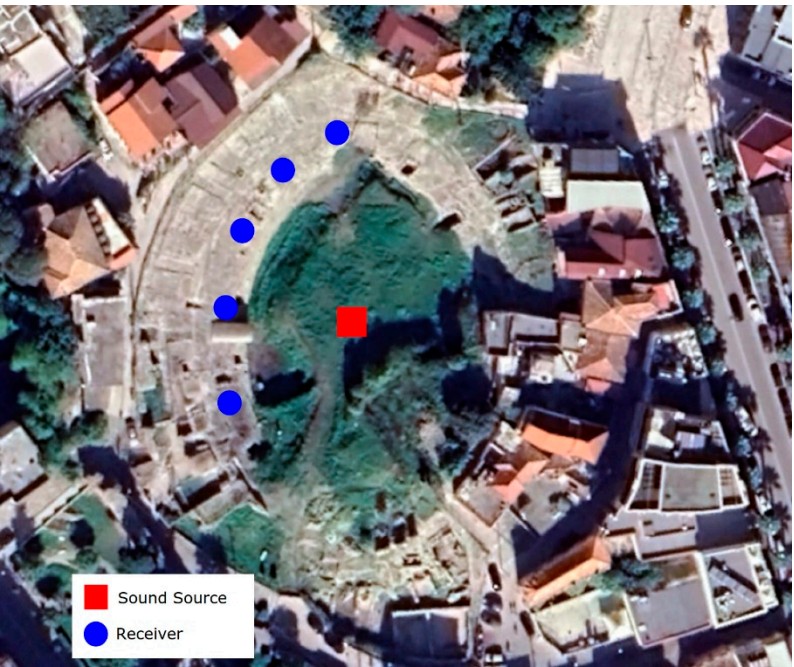

**Figure 4.** Scheme of the equipment location used during the acoustic measurements.

All acoustic measurements were carried out in unoccupied conditions. A brief description of the selected acoustic parameters is given below:

- Early decay time (EDT) is the time corresponding to 10 dB from the decay curve, starting from the impulse level measured after the interruption of the noise source. The EDT is particularly sensitive to the receiver position inside the audience area [17];
- Similar to EDT, the reverberation time ($T_{20}$) is defined as the time required for the sound pressure level to drop by 60 dB after the noise impulse [18,19]. The value of this parameter is in the function of the room volume and the amount of absorption inside a room. For open-air theaters, the sky, substituting the ceiling, is considered to be totally absorbing;
- Clarity index ($C_{50}$) is a characteristic parameter related to speech perception in a hall. It is given by the ratio between the early energy that reaches the listener within the first 50 ms (including the early reflections) and the energy arriving in the following instants [20];
- Definition ($D_{50}$) is a parameter related to the intelligibility of speech and music understanding. It is obtained from the ratio of the sound energy reaching the receiver in the first 50 ms and the entire excitation signal emitted by the sound source [21]. The early reflections added to the direct sound contribute to a positive speech understanding.

Recorded impulse responses (IRs) were processed using Audition 3.0 software, provided with the plug-in suite for analysis of the main acoustic parameters. The results were averaged over all receiver positions in the octave bands between 125 Hz and 4 kHz. The measured background noise was equal to an equivalent sound pressure level of $L_{Aeq}$ 52 dBA over a 30 min duration, mainly due to road traffic noise, which did not affect the acoustic measurements.

## 6. Digital Model

Based on the assumptions explained in a previous section, a digital model was created using AutoCAD software to reproduce the characteristics of the amphitheater, including

the number of steps of the different *maeniania* of the *cavea*. The numerical model has the following characteristics:

- Number of surfaces: 10,816;
- Total surface area: 71,388 m$^2$.

After exporting the model in the DXF format, it was utilized in Ramsete software for the acoustic simulations [22,23]. Ramsete software is based on pyramid tracing algorithm and geometric acoustics, capable of solving the sound propagation inside room. It takes into account specular and diffuse reflections over sound absorbing surfaces. The principle of spreading is based on Snell's law for light ray incident on a plane surface; the incident ray is reflected by the surface with an equal angle, and the energy of the reflected ray is reduced by a percentage according to the absorption coefficient assigned to the surface. Ramsete software is also equipped with a tool for auralization, the results of the simulation can be converted into an impulse response, to be selected from the monoaural output file to binaural and up to the 7th order Ambisonics.

The software for architectural acoustics based on raytracing evaluates the reflection of the sound in terms of energy, which simplifies the more complex procedure of the reality that involves the amplitude and phase of the soundwave [24]. Another limitation of the acoustic simulation is the discretization of the surfaces because the calculation is based on the intersection of a plane surface with a straight line (incident ray) which is therefore mirrored with an equal and opposite angle. For architectural simplification reasons, the curved surfaces are approximated to plane surfaces. The greater the number of plane surfaces used to discretize a curved surface, the greater the calculation time and the complexity of the model. In terms of scattering, the steps on the *cavea* should be considered diffusing surfaces and this phenomenon can be evaluated by the scattering coefficients [25].

A total number of 176 virtual omnidirectional microphones were created for the acoustic simulations, which were uniformly distributed over the seating area at a height of 1.3 m above the relative finish floor, keeping a constant distance of 8 m between each other on a regular squared grid. The virtual omnidirectional source was instead located in the arena. Figure 5 shows a view of the digital reproduction of the amphitheater of Durres with the characteristics of the original shape based on the assumptions described above. An external box was drawn around the amphitheater, characterized by the absorbing coefficients equal to 1 for all octaves.

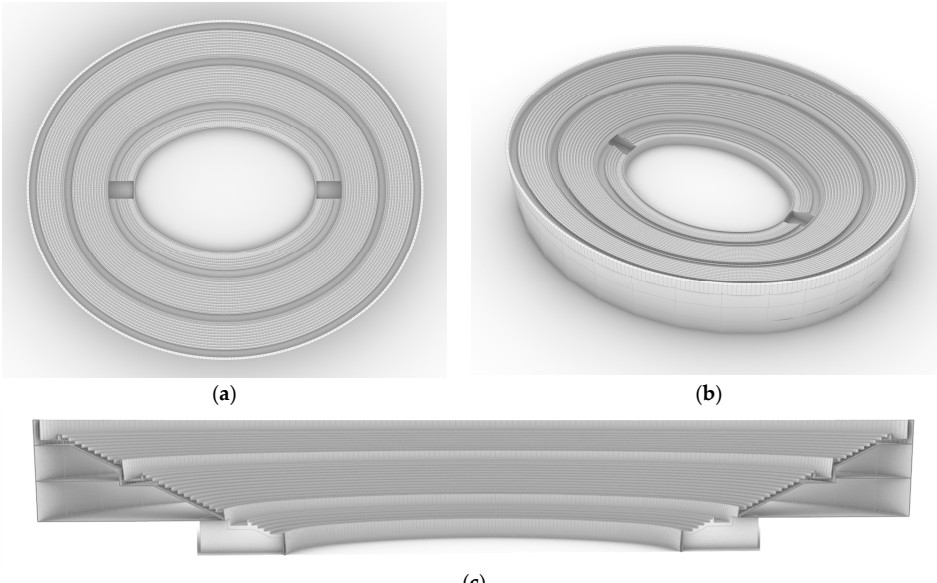

(a)   (b)

(c)

**Figure 5.** Digital model of the Roman amphitheater of Durres: (**a**) plan; (**b**) perspective; (**c**) longitudinal section. Provision courtesy of the model from Arch. Arsim Murseli, UBT University.

### 7. Acoustic Simulations

Before calculating the acoustic parameters of the original shape, absorption and scattering coefficients were assigned to all 3D faces of the digital model based on research conducted by the authors on a variety of Roman theaters and amphitheaters [26–30], as well as on their wide experience using acoustic simulation techniques [8,31,32].

It is good practice to carry out an acoustic calibration of an original reconstruction for a digital reproduction representing an existing condition [33]. This procedure was intentionally avoided in this case because the current conditions of the archaeological site are affected by the consistent presence of extraneous buildings that can distort the calibration process. Instead, coefficients measured with a laser doppler vibrometer on materials from another Roman theater were used. This technique has been widely described in previous literature [34]. On this basis, Table 1 summarizes the absorption coefficients and scattering assigned to the surfaces of the numerical model after the calibration process.

**Table 1.** Surface, absorption and scattering (bold and italics) coefficients of the materials considered in the model of the Roman amphitheater of Durres.

| Materials | Area (m$^2$) | Octave Frequency Bands (Hz) | | | | | |
|---|---|---|---|---|---|---|---|
| | | **125** | **250** | **500** | **1 k** | **2 k** | **4 k** |
| Stone—*Cavea* | 16,833 | 0.01 | 0.01 | 0.01 | 0.01 | 0.01 | 0.01 |
| | | *0.70* | *0.55* | *0.40* | *0.20* | *0.10* | *0.05* |
| Timber wood—Doors | 24 | 0.19 | 0.32 | 0.10 | 0.02 | 0.02 | 0.02 |
| | | *0.16* | *0.10* | *0.05* | *0.01* | *0.01* | *0.01* |
| Sand—Arena floor | 1413 | 0.60 | 0.7 | 0.70 | 0.50 | 0.43 | 0.30 |
| | | *0.23* | *0.20* | *0.13* | *0.10* | *0.05* | *0.01* |
| Audience | 2118 | 0.51 | 0.64 | 0.65 | 0.59 | 0.58 | 0.52 |
| | | *0.30* | *0.25* | *0.15* | *0.08* | *0.01* | *0.01* |
| External box | 53,118 | 1.00 | 1.00 | 1.00 | 1.00 | 1.00 | 1.00 |
| | | *0.01* | *0.01* | *0.01* | *0.01* | *0.01* | *0.01* |

The plots reported in Figure 6 show the simulated results compared with the measured values, which are to be considered as the average of all of the receivers. The simulations were carried out without and with an audience at full capacity over a bandwidth between 125 Hz and 4 kHz.

Figure 6a shows that the EDT values measured in the amphitheater fluctuated around 1.0 s, moderately below the optimal range that was perfectly matched by the simulated results with an audience. The simulated results without an audience were above the optimal range limit. This means that in the Roman period, the early reflections supported the sound source inside the amphitheater when the audience was present [35].

In terms of reverberation, Figure 6b shows that the measured values are around 1.35 s, which is slightly high for an outdoor environment [35]. It was found that the presence of posthumous constructions contributed to an increasing number of reflections bouncing off the vertical hard surfaces. The reverberation time of the original structure when fully occupied was approximately 2.0 s, which is very similar to the simulated results carried out by the authors inside the Roman amphitheater of Avella in its original shape [36,37]. Although this result seems high for an open-air theater, it should be considered as a function of its volume. Without an audience, the simulated results shifted upward by 0.6 s, highlighting how the amount of surface area related to an audience can consistently improve the acoustic response, even in open-air construction [38].

Figure 6c shows that the definition is very similar for all three scenarios, fluctuating around 0.75, meaning that the conditions were and remain good for speech performance.

In terms of speech clarity, it is good practice to show the results along with acoustic maps showing the variation across the seating area [39]. In this case, Figure 7 shows the simulated results of $C_{50}$ in the amphitheater of Durres at 1 kHz, since it is an octave band suitable for assessing both the male and female voice. Figure 7 indicates that no great

difference was found between the two scenarios and that the highest values were found at the position of the *vomitoria* and also at the top of the *summa cavea*, along the major axis. The values that, instead, were closer to the optimal range were found along the minor axis, fluctuating between 0 dB and 2 dB.

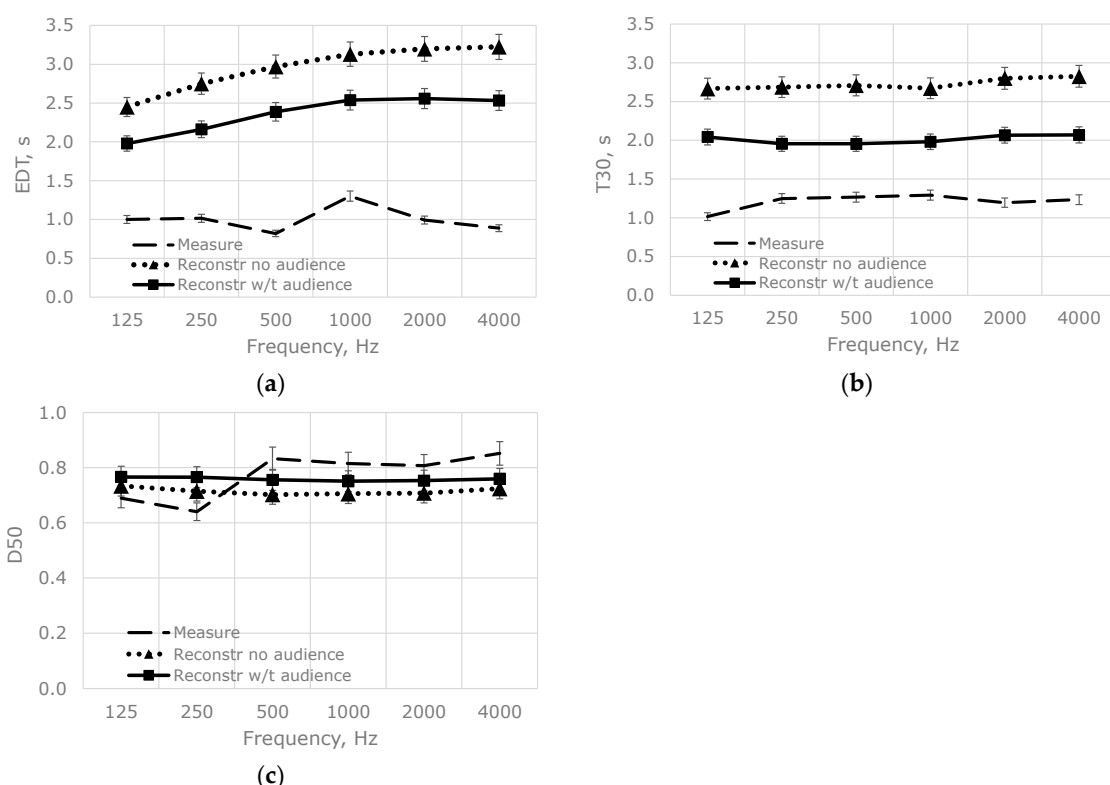

**Figure 6.** Comparison between the measured and simulated values related to (**a**) EDT; (**b**) reverberation time; (**c**) definition.

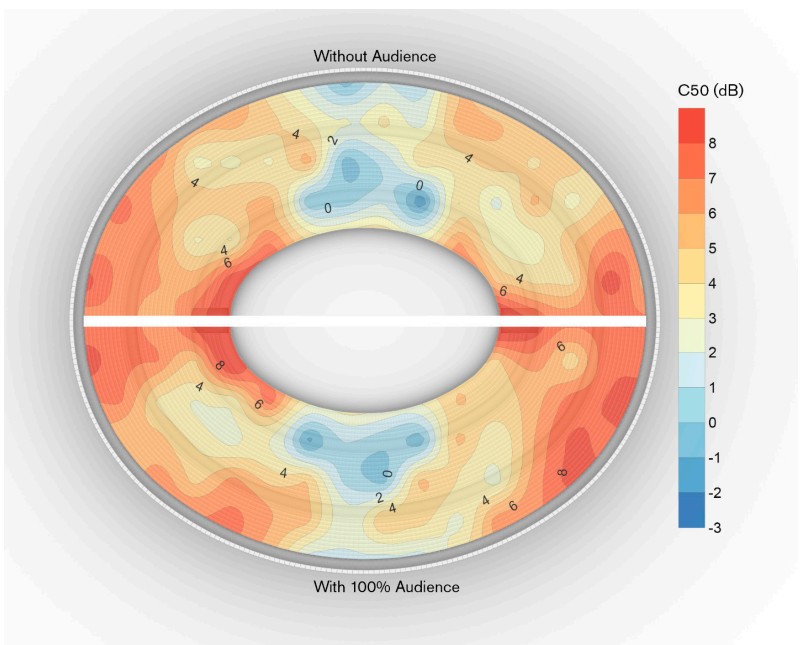

**Figure 7.** Acoustic map of speech clarity: simulated results related to a full occupancy and without any audience.

In addition to an analysis of the acoustic parameters, the impulse responses (IRs) gathered from the simulations were studied by taking into consideration three different position across the sitting area:

- In the arena;
- In the *ima cavea*;
- In the *summa cavea*.

The three IRs were plotted on the same graph, as shown in Figure 8 in the time domain, as expressed on the x-axis.

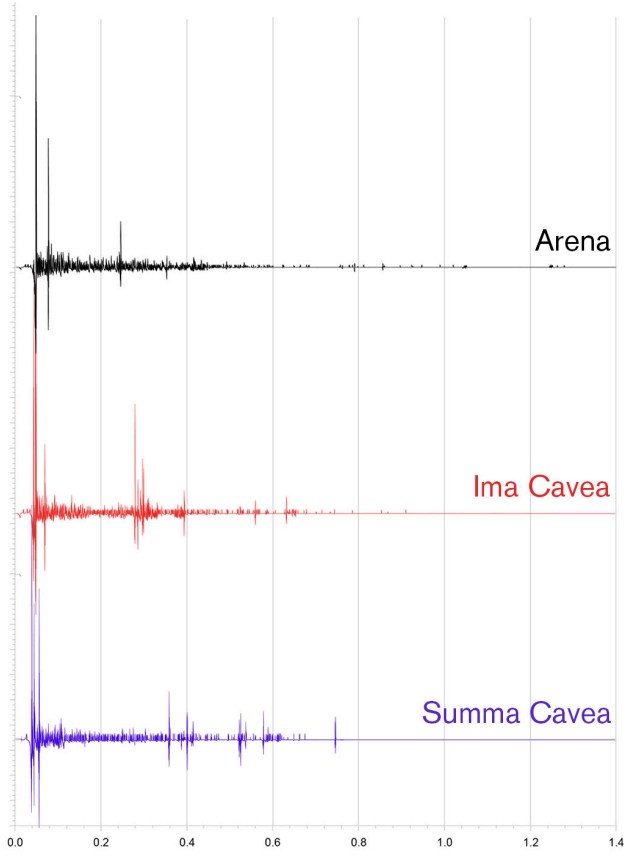

**Figure 8.** Impulse responses obtained from the acoustic simulations in three specific points: arena, *ima cavea*, and *summa cavea*.

Figure 8 shows that in the arena, the reflection coming from the wall at the perimeter of the arena floor was very strong, as is visible from the second peak detected after the direct sound.

The sound reflection is generated by the boundary walls between the arena and the *cavea*. This wall of separation was built to protect the spectators from the dangers that could arise during the fights. The multiple reflections of the sound on the wall of the arena are distinctly perceivable [37] and generate the flutter echo effect, as the energy is trapped between two parallel surfaces.

This reflection was less strong in the *ima cavea*, although another reflection was more visible after 20 ms.

The reflection after 20 ms on the graph related to the *ima cavea* was because of geometric reasons, since the shape was symmetrical, and the strong reflection visible in the *ima cavea*, which came from the steps located on the diametrically opposite side of the receiver point, was taken into consideration.

The graph related to the IR of the *summa cavea* indicated only a strong reflection soon after the direct sound but not those coming from the *balteum*. The small peaks 0.4 s and

0.6 s after the direct sound were related to the geometric shape, as explained previously. The flutter-echo effect is weaker for the points located in the *summa cavea*.

## 8. Conclusions

In the Imperial Age every city of the Roman empire had an amphitheater, the size of which varied according to the importance of the city. Gladiators' shows were held inside amphitheaters and were highly appreciated by Romans. The decline and abandonment of the amphitheaters was a consequence of the advent of the Catholicism because considered cruel and anti-ethic. Therefore, some amphitheaters from a place of entertainment have been transformed into cemeteries or catacombs.

The realization of the amphitheaters was significant during the Imperial Age with a total number of 300 surviving to our days, located throughout Europe, North Africa and Asia. Amphitheaters in modern times have become famous for many movies and television series that have been set. Most of the time, the reconstruction of this type of building is often based on the necessity of hosting live shows or also adapted to open museums.

This paper showed and discussed the main acoustic parameters related to the original reconstruction of the Roman amphitheater of Durres. The simulated results would be more suitable for a live musical performance than the existing condition, which were affected by the presence of residencies invading the archaeological space. The outcomes of this study can be used for virtual reality or by cinematography businesses for the reconstruction of ancient periods in their films. These results can support their audio track reproductions.

Future research studies will focus on the comparison of the acoustics related to different amphitheaters rebuilt in their original form, composed of different sizes and dimensions.

**Author Contributions:** Conceptualization, S.S., G.I., A.T. and A.B.; methodology, G.I. and A.B.; software, A.B. and G.I.; validation, A.B. and S.S.; formal analysis, A.B.; investigation, S.S., A.T. and G.I.; resources, S.S.; data curation, A.B.; writing—original draft preparation, G.I., A.B. and S.S.; writing—review and editing, A.B.; visualization, A.B.; supervision, G.I. and S.S.; project administration, G.I., A.B., S.S. and A.T.; funding acquisition, S.S. and A.B. All authors have read and agreed to the published version of the manuscript.

**Funding:** This research received no external funding.

**Data Availability Statement:** MDPI Research Data Policies.

**Acknowledgments:** A special thanks goes to Arsim Murseli for the drawings and collaboration on the graphics of this manuscript.

**Conflicts of Interest:** The authors declare no conflict of interest.

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
