# Peer review of "Insight Discovery of the Roman Amphitheater of Durres: Reconstruction of the Acoustic Features to Its Original Shape"

_buildings, doi:10.3390/buildings13071843_

Round 1

Reviewer 1 Report

It would be worthwhile to provide the sound pressure level, since the background sound of 52 dB is quite high, the lowest measurement point after the sound drop should not be lower than 62 dB.

Author Response

It would be worthwhile to provide the sound pressure level, since the background sound of 52 dB is quite high, the lowest measurement point after the sound drop should not be lower than 62 dB.

Dear reviewer, the background Laeq 52 dB was measured over 30 minutes duration. The survey was affected by traffic noise but the impulse response was taken during a gap where no vehicles pass-by could affect the decay, otherwise the computation in the post processing would give null results.

Thank you

Reviewer 2 Report

This is an interesting archaeoacoustic study, and you have a tricky problem due to the current state of the site. However, you don’t spend enough time addressing the key components on which your study succeeds or fails - specifically 

1) what is the overall layout of the site as measured today? 

2) can you compare specific source-receiver combinations, and look at the echogram to infer specific reflections that would be removed (or added) in your historical simulation?

3) what are the assumptions behind the absorption coefficients used? How were they measured, and what is the uncertainty of these values?

4) How does the acoustic simulation algorithm work? You didn’t say anything about it here. Based on your results, it seems like it could potentially have a significant impact on your final simulations. Given all the uncertainty that there is in historical modeling, you need to have the assumptions of the simulation engine front and center, such that any remaining uncertainties in the final output is obvious to the reader.

Broad issues, page by page:

P2: I have never heard this definition of an oval before - I have always heard that an oval is more vaguely defined, as a smooth elongated curve with at least one axis of symmetry, such that ellipses are a subset of ovals. Can you provide a citation to this definition? Later on you use “polycentric oval” so I wonder if that is the more precise term. Also, maybe a diagram would help to explain exactly what you mean here since it seems to be important to your analysis.

P2: “The development of Christianity” is probably better phrased as “the growth of Christianity” or something similar. Christianity was “developed” prior to the 4th century, after all.

P2: can you define what the “balteum” is? You didn’t mention it above.

P2 (and throughout) - your dates all have “AC” after them, which I assume you are using for “After Christ,” which is confusing because some historians used AC for “Ante Christum” or essentially meaning BC. Most journals would use BC/AD for dates, or else BCE/CE. But I think AC is just going to confuse people.

P3: could you define “summa cavea” ?

P3: “It was tempting to dig” - when? For whom? Again, the passive voice is a bit unclear.

P5: Regarding the use of firecrackers and ISO-3382 requirements, I think you should include Papadakis, N. M. and Stavroulakis, G. E. (2019) ‘Review of acoustic sources alternatives to a dodecahedron speaker’, Applied Sciences (Switzerland), 9(18). doi: 10.3390/app9183705.

…which contains this conclusion: 

“In conclusion, the firecracker as a sound source can potentially meet the ISO 3382-1 requirements concerning omnidirectionality. Research needs to be carried out according the ISO 3382-1 measurement procedure to certify that the firecracker can cover this requirement. Concerning the sound pressure levels requirements of ISO 3382-1, as stated in the introduction, they need to be at least 45 dB above the background level in the corresponding frequency band. Whether this condition is met depends on the firecracker used and the sound level that it creates in the low-frequency range, as well as the background noise in this frequency range.”

Do you have more information on the background noise levels at low frequencies? Do you know that your firecrackers were 45 dB above at those levels (or 35 dB if you’re only using T20)?

Otherwise, your background literature review seems pretty good for studies on amphitheaters - the only thing that I would like a bit more information on is the effect mentioned in Declercq & Dekeyser (2007) on the filtering effect of corrugated seating at Epidaurus: based on your model do you think you would observe a similar effect at Durres?

P6: in your definitions of C50 and D50, you call these “correlations” but I think “ratio” or “dB ratio” would be more informative if readers don’t already know what these are.

P7: It is common in GA modeling to reduce or simplify a CAD model for the purposes/wavelengths of acoustic simulation, as often more surfaces do not necessarily lead to a more accurate result. Are the 10,816 surfaces the amount that came out of the CAD model, or is that after simplification, or was there no simplification?

P8: if the absorption coefficients here were measured in a previous publication, could you provide citations to where each measurement comes from? Also, is this an approximation of normal-incidence absorption coefficient or the random-incidence absorption coefficient? 

P8: how does Ramsete handle scattering? Is the coefficient given here extrapolated over all octave bands, or is there a nonlinear formula used for estimating scattering at each frequency?

P8: These curves are the average values over all of your virtual microphones? Please report the standard deviations as error bars 

P8: You use a lot of terms like “the early reflections supported” or “the reverberation time of the original structure was about 2.0s” but these are just the results of your simulations! What are the sources of uncertainty in these simulations? Is the pyramid-tracing algorithm stochastic or deterministic? If it’s stochastic, did you run it multiple times and average the outputs? 

P8: Also, your simulated T30 results look very fishy! Not only is it very high for an outdoor space (as you mention), but why are they the same across all octave bands? Even in an outdoor space, if the model is accounting for air absorption, you should get a noticeable reduction in T30 by 4000 Hz at least. What are the temperature and humidity in the model?

More generally, it would make sense if we are having a highly non diffuse field, where a series of lateral echoes are dominating, even as the ceiling is 100% absorbent. It can be the case that T20 or even T30 can get somewhat bigger due to a double-slope decay - can you provide the entire simulated decay curve to see whether that’s the case? 

P9: Figure 8: are you displaying values for C80 for an omnidirectional source, or for a human voice directivity pattern? If the latter, then it would matter whether it was measured for a male or female; if the former, then it’s not necessarily the best simulation of how a voice would have been heard. 

You have a few minor grammatical or spelling errors:

Abstract: “The results show that the results…” is a bit awkward phrasing and should be re-written

P1: “it is a hypothesis that the amphitheaters…” - passive voice: who hypothesized this? Are you hypothesizing it right now? Please be clear about this.

P2: “Middle Age” should be “Middle Ages”

P2: “intension” should be “intention”

P3: “under posthumous constructions” makes it sound like it’s still being built upon. I think “posthumous buildings” would be clearer, or just “beneath later constructions.”

P9: “mayor” axis should be “major” axis

P9: “The simulated results to be” should be “The simulated results would be”

Author Response

This is an interesting archaeoacoustic study, and you have a tricky problem due to the current state of the site. However, you don’t spend enough time addressing the key components on which your study succeeds or fails - specifically 

1) what is the overall layout of the site as measured today? 

Dear reviewer, the conditions of the site have been added in Figure 4

2) can you compare specific source-receiver combinations, and look at the echogram to infer specific reflections that would be removed (or added) in your historical simulation?

Dear reviewer, the source-receiver combination have been compared in terms of IRs, in this way it is possible to see the reflection based on the position of the receiver, as indicated in figure 9.

3) what are the assumptions behind the absorption coefficients used? How were they measured, and what is the uncertainty of these values?

Dear reviewer, The absorption coefficients have been measured in another Roman theatre, well explained in ref [30]. However, the coefficients reported in the table are the results of the calibration process, which is a procedure undertaken to tune the digital model. This is explained in the paper.

4) How does the acoustic simulation algorithm work? You didn’t say anything about it here. Based on your results, it seems like it could potentially have a significant impact on your final simulations. Given all the uncertainty that there is in historical modeling, you need to have the assumptions of the simulation engine front and center, such that any remaining uncertainties in the final output is obvious to the reader.

Dear reviewer, the acoustic simulations follow the equations of the sound spreading in free-field or room acoustics, wherever applicable. This is a computation done by the ramsete software (http://www.ramsete.com/).

Broad issues, page by page:

P2: I have never heard this definition of an oval before - I have always heard that an oval is more vaguely defined, as a smooth elongated curve with at least one axis of symmetry, such that ellipses are a subset of ovals. Can you provide a citation to this definition? Later on you use “polycentric oval” so I wonder if that is the more precise term. Also, maybe a diagram would help to explain exactly what you mean here since it seems to be important to your analysis.

Dear reviewer, oval is a technical term used widely in geometry. See this article relative to an “oval dome” (https://www.researchgate.net/publication/227327355_Oval_Domes_History_Geometry_and_Mechanics). Moreover, the discussion between ovals and ellipses particularly applied to roman amphitheater is extensively explained in the new ref [13]. Please, be documented.

P2: “The development of Christianity” is probably better phrased as “the growth of Christianity” or something similar. Christianity was “developed” prior to the 4th century, after all.

Dear reviewer, growth has been changed with development.

P2: can you define what the “balteum” is? You didn’t mention it above.

Dear reviewer, this has been added.

P2 (and throughout) - your dates all have “AC” after them, which I assume you are using for “After Christ,” which is confusing because some historians used AC for “Ante Christum” or essentially meaning BC. Most journals would use BC/AD for dates, or else BCE/CE. But I think AC is just going to confuse people.

Dear reviewer, AC has been deleted everywhere. 

P3: could you define “summa cavea” ?

Dear reviewer, this has been added.

P3: “It was tempting to dig” - when? For whom? Again, the passive voice is a bit unclear.

Dear reviewer, the sentence has been changed.

P5: Regarding the use of firecrackers and ISO-3382 requirements, I think you should include Papadakis, N. M. and Stavroulakis, G. E. (2019) ‘Review of acoustic sources alternatives to a dodecahedron speaker’, Applied Sciences (Switzerland), 9(18). doi: 10.3390/app9183705.

…which contains this conclusion: 

“In conclusion, the firecracker as a sound source can potentially meet the ISO 3382-1 requirements concerning omnidirectionality. Research needs to be carried out according the ISO 3382-1 measurement procedure to certify that the firecracker can cover this requirement. Concerning the sound pressure levels requirements of ISO 3382-1, as stated in the introduction, they need to be at least 45 dB above the background level in the corresponding frequency band. Whether this condition is met depends on the firecracker used and the sound level that it creates in the low-frequency range, as well as the background noise in this frequency range.”

Do you have more information on the background noise levels at low frequencies? Do you know that your firecrackers were 45 dB above at those levels (or 35 dB if you’re only using T20)?

Dear reviewer, this ref has been added. The authors are aware of what stated in ISO 3382, and by using the plugin Aurora in Audition software, as explained in the paper, the values are obtained, meaning that there is sufficient time for the sound energy to decay, otherwise the result was null.

Otherwise, your background literature review seems pretty good for studies on amphitheaters - the only thing that I would like a bit more information on is the effect mentioned in Declercq & Dekeyser (2007) on the filtering effect of corrugated seating at Epidaurus: based on your model do you think you would observe a similar effect at Durres?

Dear reviewer, the condition of the cavea nowadays is highly ruined and nobody knows what the real marble sheet could be used for seating. However, in the acoustic simulations a scattering coefficient of 0.55 on the low frequency up to 500 Hz has been assumed, and the relative results are shown in the graphs of figure 6. Many could be hypotheses, and many can be the numbers that can be used for the simulations. The corrugation effect surely influences the diffusiveness of sound inside the open-air theatre. Due to the huge lack of information, it has been avoided on purpose to talk specifically about this since only hypotheses can be made on it.

P6: in your definitions of C50 and D50, you call these “correlations” but I think “ratio” or “dB ratio” would be more informative if readers don’t already know what these are.

Dear reviewer, this has been done.

P7: It is common in GA modeling to reduce or simplify a CAD model for the purposes/wavelengths of acoustic simulation, as often more surfaces do not necessarily lead to a more accurate result. Are the 10,816 surfaces the amount that came out of the CAD model, or is that after simplification, or was there no simplification?

Dear reviewer, 10,816 is the number of surfaces counted in Ramsete, that were necessary to build the most accurate as possible the model as shown in Figure 5. Since the software does not read curve surfaces, any curvature has been approximated with flat surfaces. This is the main core of simplification.

P8: if the absorption coefficients here were measured in a previous publication, could you provide citations to where each measurement comes from? Also, is this an approximation of normal-incidence absorption coefficient or the random-incidence absorption coefficient? 

Dear reviewer, the reference is [32] which explains all the process for carrying on measurements of materials inside the roman theatre of tyndaris.

P8: how does Ramsete handle scattering? Is the coefficient given here extrapolated over all octave bands, or is there a nonlinear formula used for estimating scattering at each frequency?

Dear reviewer, for simplification, only one octave was indicated in the table, but yes ramsete can take in consideration of all the frequencies, from 31.5 Hz to 16 kHz. Updates on table 1 have been made.

P8: These curves are the average values over all of your virtual microphones? Please report the standard deviations as error bars.

Dear reviewer, error bars have been added.

P8: You use a lot of terms like “the early reflections supported” or “the reverberation time of the original structure was about 2.0s” but these are just the results of your simulations! What are the sources of uncertainty in these simulations? Is the pyramid-tracing algorithm stochastic or deterministic? If it’s stochastic, did you run it multiple times and average the outputs? 

Yes, they are simulated results. the pyramid-tracing algorithm used in ramsete is deterministic; the uncertainty depends only on the values of abs and scattering coefficients inserted other than to the model definition, meaning that if the model is highly detailed the advantage of more accurate results is against the computation time.

P8: Also, your simulated T30 results look very fishy! Not only is it very high for an outdoor space (as you mention), but why are they the same across all octave bands? Even in an outdoor space, if the model is accounting for air absorption, you should get a noticeable reduction in T30 by 4000 Hz at least. What are the temperature and humidity in the model?

Fishy?? Sorry but this provocatory attitude is not accepted. Many reconstructions of the original shapes are based on measured results and can give simulated values similar to what found here for Durres. It has been specified that we are not taking this procedure since the model calibration, with the presence of extraneous buildings, can mislead completely the abs and scattering coefficients. The graphs highlight the difference between measured and simulated results, but it cannot be said that the simulated numbers are completely invented!

More generally, it would make sense if we are having a highly non diffuse field, where a series of lateral echoes are dominating, even as the ceiling is 100% absorbent. It can be the case that T20 or even T30 can get somewhat bigger due to a double-slope decay - can you provide the entire simulated decay curve to see whether that’s the case? 

Dear reviewer, the simulated decay curves do not have double slope decay. It has been added to the manuscript that a box was drawn around the amphitheater, characterized by alpha 1 over all the spectrum and to all the 6 faces.

P9: Figure 8: are you displaying values for C80 for an omnidirectional source, or for a human voice directivity pattern? If the latter, then it would matter whether it was measured for a male or female; if the former, then it’s not necessarily the best simulation of how a voice would have been heard. 

Dear reviewer, the simulations have been carried out with an omnidirectional source, with a power level of 110 dB flat across the bandwidth, which can be the worst scenario by considering all the possible directivity. If the source would have been a male or female voice then the results would be not accurate, since the power level of a human voice would not reach in this case the latest rows of seats. In any case, the directivity here is not applicable at all since the gladiators would be moving all around instead of thinking a speaker fixed in a specific position and talking to an audience, which is the case of an opera theatre… 

Comments on the Quality of English Language

You have a few minor grammatical or spelling errors:

Abstract: “The results show that the results…” is a bit awkward phrasing and should be re-written

This has been corrected.

P1: “it is a hypothesis that the amphitheaters…” - passive voice: who hypothesized this? Are you hypothesizing it right now? Please be clear about this.

This has been corrected.

P2: “Middle Age” should be “Middle Ages”

This has been corrected.

P2: “intension” should be “intention”

This has been corrected.

P3: “under posthumous constructions” makes it sound like it’s still being built upon. I think “posthumous buildings” would be clearer, or just “beneath later constructions.”

This has been corrected.

P9: “mayor” axis should be “major” axis

This has been corrected.

P9: “The simulated results to be” should be “The simulated results would be”

This has been corrected.

Reviewer 3 Report

This paper conducted a measurement and simulation of the acoustical characteristics of the Roman amphitheaters. This work is novelty and useful to help us understand the acoustical characteristics of the historical architecture. Some revisions should be done before it can be published in the journal Building.

1. A complete finite element or boundary element model for acoustic simulation should be presented and supplemented in the paper, and all detailed acoustic simulation information such as the size of the acoustic grid, boundary conditions, calculation frequency, etc. should be introduced in detail.

Author Response

A complete finite element or boundary element model for acoustic simulation should be presented and supplemented in the paper, and all detailed acoustic simulation information such as the size of the acoustic grid, boundary conditions, calculation frequency, etc. should be introduced in detail.

Dear reviewer, a finite element model is not mandatory to be in place. Different are the software used for acoustic simulations and the authors just used one among different available on the market. Information of the grid used to deploy the microphone has been added, along with the calculation frequency and the boundary conditions.

Thank you